# NavBench: Probing Multimodal Large Language Models for Embodied Navigation

Yanyuan Qiao[1]    Haodong Hong[2][3]    Wenqi Lyu[4]    Dong An[5]    Siqi Zhang[6]
Yutong Xie[5]    Xinyu Wang[4]    Qi Wu[4]*

[1]Swiss Federal Institute of Technology Lausanne (EPFL)
[2]The University of Queensland [3]CSIRO Data61 [4]The University of Adelaide
[5]Mohamed bin Zayed University of Artificial Intelligence [6]Tongji University

🌐 Project Website

## Abstract

Multimodal Large Language Models (MLLMs) have demonstrated strong generalization in vision-language tasks, yet their ability to understand and act within embodied environments remains underexplored. We present NavBench, a benchmark to evaluate the embodied navigation capabilities of MLLMs under zero-shot settings. NavBench consists of two components: (1) navigation comprehension, assessed through three cognitively grounded tasks including global instruction alignment, temporal progress estimation, and local observation-action reasoning, covering 3,200 question-answer pairs; and (2) step-by-step execution in 432 episodes across 72 indoor scenes, stratified by spatial, cognitive, and execution complexity. To support real-world deployment, we introduce a pipeline that converts MLLMs' outputs into robotic actions. We evaluate both proprietary and open-source models, finding that GPT-4o performs well across tasks, while lighter open-source models succeed in simpler cases. Results also show that models with higher comprehension scores tend to achieve better execution performance. Providing map-based context improves decision accuracy, especially in medium-difficulty scenarios. However, most models struggle with temporal understanding, particularly in estimating progress during navigation, which may pose a key challenge.

## 1 Introduction

Multimodal Large Language Models (MLLMs) [1, 2, 3] have achieved impressive performance across a wide range of vision-language tasks, demonstrating strong cross-modal reasoning and zero-shot generalization. These models excel at answering visual questions [4], interpreting videos [5], and performing complex multimodal reasoning [6]. As their capabilities expand, a central question emerges: do these models truly understand how to act in the physical world, or are they simply adept at processing static inputs?

Recent work has begun to explore MLLMs' potential in embodied tasks by evaluating their spatial reasoning in 3D environments [7, 8]. However, these tasks primarily focus on perception and passive scene understanding, without assessing the model's ability to make decisions or take actions. In comparison, navigation is a core embodied task that involves interpreting natural language instructions, analyzing visual observations, and making a sequence of decisions to reach a goal. Although navigation plays a crucial role in real-world applications, it remains relatively underexplored in the context of MLLMs. Traditional embodied navigation benchmarks, such as Room-to-Room (R2R) [9]

---

*Corresponding author

39th Conference on Neural Information Processing Systems (NeurIPS 2025).

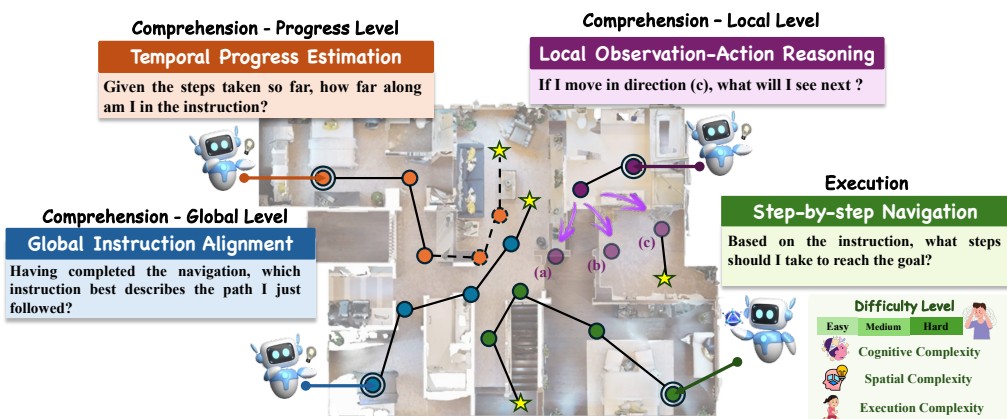

Figure 1: NavBench evaluates MLLMs across three comprehension tasks and a step-by-step execution task, assessing their ability to understand navigation behavior, track progress, reason about observation and action, and act accordingly. The step-by-step navigation is assessed from different difficulty levels, which is defined by cognitive, spatial, and execution complexity.

and ObjectNav [10], were developed prior to the emergence of foundation models. These benchmarks rely on task-specific supervision and often reduce evaluation to final success rates, providing limited insight into whether a model genuinely understands the navigation behavior. In many cases, an agent may reach the goal by exploiting dataset biases or learning shortcuts, without correctly grounding the instruction or following the intended path.

Similar to how humans acquire embodied skills by first understanding a task and then learning to execute it, evaluating the embodied capabilities of generalist MLLMs also requires examining two fundamental aspects. First, can the model comprehend what a navigation behavior represents, such as identifying the intent behind a completed trajectory? Second, can it act autonomously to complete a navigation task, making step-by-step decisions in unfamiliar environments? Furthermore, navigation tasks in real-world environments can vary significantly in difficulty due to differences in spatial layout, instruction complexity, and required decision-making steps. For example, navigating across multiple rooms with ambiguous instructions poses greater challenges than following simple step-by-step commands in a single hallway. However, most existing benchmarks treat all navigation episodes equally difficult, failing to capture this essential variation.

To fill these gaps, we introduce **NavBench**, a benchmark designed to systematically evaluate MLLMs in embodied navigation under zero-shot settings. NavBench decomposes the evaluation into two complementary components: *Navigation Comprehension*, which assesses whether a model understands and aligns with intended navigation behavior, and *Navigation Execution*, which evaluates the model's ability to make accurate step-by-step decisions. To reflect real-world variability, NavBench incorporates a fine-grained difficulty classification based on spatial, cognitive, and execution complexity. In addition, it provides a deployable real-world navigation pipeline to bridge the gap between simulation and practical embodiment.

**First**, for navigation behavior comprehension, inspired by cognitive studies of human spatial reasoning [11], NavBench introduces three fine-grained evaluation tasks designed to assess distinct reasoning capabilities at three levels: global, progress, and local. It includes 3,200 question-answer pairs. Specifically, *Global Instruction Alignment* evaluates the model's ability to match a given trajectory with the most appropriate instruction. The candidate instructions are designed with subtle semantic differences, such as variations in directional cues and landmark descriptions, to encourage genuine spatial reasoning. *Temporal Progress Estimation* measures temporal-contextual awareness by requiring the model to infer progress within multi-step instructions based on a partial trajectory. *Local Observation-Action Inference* evaluates the model's ability to reason about the spatial consequences of individual actions by either predicting the future observation given an action or identifying the action that caused a visual transition. Together, these tasks provide a comprehensive framework for assessing global semantic reasoning, temporal understanding, and local spatial inference in navigation.

**Second**, NavBench introduces a fine-grained difficulty classification with three levels: easy, medium, and hard, based on cognitive, spatial, and execution complexity. This allows detailed analysis of

models' generalization and decision-making performance across varying levels of difficulty. The benchmark includes 432 navigation cases across 72 scenes.

**Finally**, to bridge the gap between simulator-based evaluation and real-world deployment, we design a practical navigation pipeline that connects MLLM outputs to executable actions on real robots. This pipeline includes a waypoint selection module, an MLLM-based navigator, and a low-level controller, demonstrating the deployability of our framework in physical environments.

We evaluate both closed-source and open-source MLLMs on NavBench. While GPT-4o currently achieves the best overall performance, we observe that lightweight models such as Qwen2.5-VL-7B are capable of reliably completing easy navigation tasks. Notably, this trend is also reflected in our real-world deployment experiments, suggesting that NavBench may serve as a practical tool for analyzing the embodied capabilities of both general and resource-efficient MLLMs. Furthermore, our results suggest several notable trends: (1) comprehension and execution abilities appear to be closely related, (2) temporal reasoning may pose a persistent challenge for current models, and (3) compact open-source models can, under certain conditions, approach the performance of proprietary ones, indicating their potential utility in practical settings.

In summary, our main contributions are as follows: (1) We introduce *NavBench*, a benchmark for evaluating MLLMs in embodied navigation under zero-shot settings. (2) We decompose the evaluation into two components: *Navigation Comprehension*, with tasks targeting spatial, temporal, and local reasoning, and *Navigation Execution*, which assesses decision-making across difficulty levels. (3) We develop a deployment pipeline that maps MLLM outputs to real-world robot actions. (4) We perform a detailed evaluation and analysis of both closed-source and open-source MLLMs, uncovering trends in their reasoning and execution performance across embodied tasks.

## 2 Related Work

**Benchmarks for MLLMs** Recent progress in Multimodal Large Language Models (MLLMs)[1, 12, 13, 14, 15] has driven the development of benchmarks assessing visual understanding and cross-modal reasoning. Early efforts such as VQA[4], GQA [16], OK-VQA [17], and TextVQA [18] focus on specific tasks like factual or commonsense question answering. More recent benchmarks including MME [19], MMBench [20], MM-Vet [21], and MathVista [6] aim for broader coverage, evaluating perception and reasoning across diverse domains. However, these mainly target static tasks and do not reflect MLLMs' ability to act in dynamic environments. To bridge this gap, some recent work has begun evaluating spatial reasoning in embodied settings. SpatialBench [7], ScanReason [22], and VSI-Bench [8] assess 3D spatial understanding using panoramas, semantic layouts, or textual scene descriptions. While insightful for embodied perception, they remain limited to passive tasks and do not assess decision-making or sequential interaction. In parallel, traditional embodied navigation benchmarks such as R2R[9], REVERIE[23], and ObjectNav [10] have long been used to test instruction-following agents. However, they were designed for fully supervised settings and mainly evaluate success rates without probing intermediate reasoning. Although REVERIE increases instruction abstraction, it retains similar path lengths and decision complexity, limiting its capacity to reveal behavioral differences. More recently, Wang *et al.* [24] proposed a fine-grained evaluation framework for instruction understanding in VLN via multiple-choice questions, offering interpretability beyond end-to-end metrics. Still, their setup is restricted to small supervised models and lacks real-world deployment and zero-shot inference. To the best of our knowledge, no existing benchmark offers a comprehensive evaluation of MLLMs in embodied navigation that jointly considers instruction understanding, sequential decision-making, difficulty stratification, and real-world transferability.

**Embodied Navigation** Embodied navigation tasks require an agent to reach a goal location within an environment, guided by a description such as an image [25, 26], object [10, 27], or natural language instruction [9, 28, 29]. Among these, language-guided navigation has attracted significant attention for its potential to facilitate intuitive human-robot interaction. Researchers have explored diverse instruction formats, including step-by-step [9, 30], dialog-based [31], and goal- or intention-oriented instructions [23, 32]. Traditional approaches train navigation policies using annotated datasets, incorporating modules to improve object relation understanding [33, 34, 35], vision-language alignment [36, 37, 38], memory [39, 40, 41], and spatial reasoning [42, 43, 44]. While effective on benchmarks, these methods often suffer from limited generalization due to dataset biases [45, 46, 47]. To mitigate this, recent work turns to MLLMs for zero-shot embodied navigation, leveraging their generalization abilities. Some use MLLMs to localize goal-relevant regions [48, 49, 50], while others

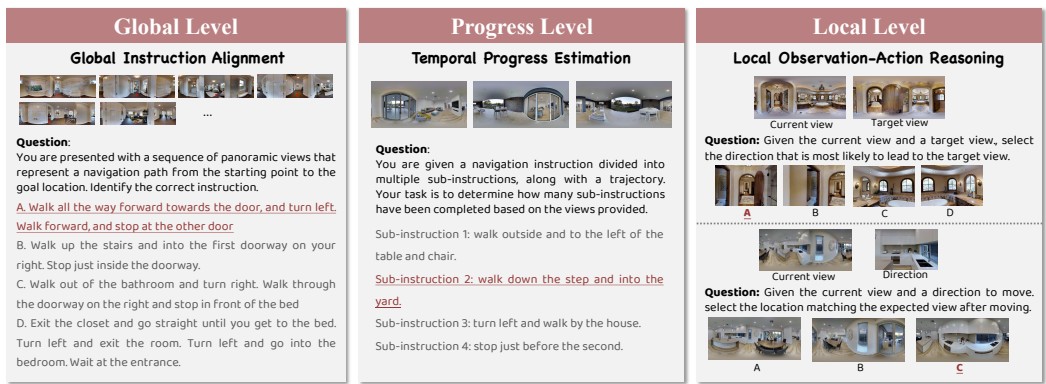

Figure 2: Illustration of the Navigation Comprehension task.

employ prompt-based guidance for instruction following [51, 52, 53, 54]. These approaches reduce reliance on task-specific training but still lack fine-grained evaluation: most benchmarks focus solely on final success rates, offering limited insight into the model's reasoning process. To address this, we introduce NavBench, a benchmark that systematically evaluates both the reasoning and execution capabilities of MLLMs in embodied navigation.

## 3 Benchmark Design

### 3.1 Task Formulation

We evaluate the navigation capabilities of MLLMs by decomposing the task into two core components: *Navigation Comprehension*, which assesses the understanding of navigation behavior, and *Navigation Execution*, which focuses on step-by-step decision making.

**Navigation Comprehension** It investigates whether the model can understand and reason about implicit navigation behaviors, including aligning instructions with trajectories, estimating progress along a plan, and predicting the spatial consequences of actions. These tasks span different reasoning levels (*global*, *progress*, and *local*) and serve as diagnostic probes for navigation understanding. Illustrations of the three comprehension tasks are shown in Figure 2[2].

- Global Level – Global Instruction Alignment: Given a navigation trajectory and several candidate instructions, the model is required to determine which instruction aligns with the executed path. This task tests the model's understanding of the overall intent and structural coherence of the navigation behavior.

- Progress Level – Temporal Progress Estimation: Provided with a partial trajectory and a list of segmented sub-instructions, the model must identify the sub-instruction that was most recently completed. This evaluates the model's capacity to monitor task progress and comprehend the temporal structure of instructions.

- Local Level – Local Observation-Action Reasoning: To evaluate the model's ability to reason about the spatial consequences of individual actions. We design two variants: (1) Future-Observation Prediction – the model observes the current view and an action, and selects the correct resulting view. (2) Future-Action Prediction – the model observes two consecutive views and must identify the action that caused the transition.

**Navigation Execution** It examines whether an MLLM can make accurate, step-by-step movement decisions in an embodied environment based on the current observation and instruction. We conduct this evaluation in a zero-shot setting [54] within the Matterport3D simulator [55], categorizing tasks into three difficulty levels (easy, medium, and hard) to assess performance. To ensure a fair and standardized evaluation protocol, we evaluate MLLMs via viewpoint selection rather than low-level action prediction (*e.g.*, turning or moving forward). This abstraction, consistent with prior embodied navigation benchmarks [9, 23], allows us to focus on high-level semantic reasoning grounded in language and vision, while avoiding the confounding variability introduced by continuous control. It also facilitates zero-shot evaluation and comparability across different models. Notably, while our

---

[2]The questions in the figure are slightly simplified for clarity and brevity.

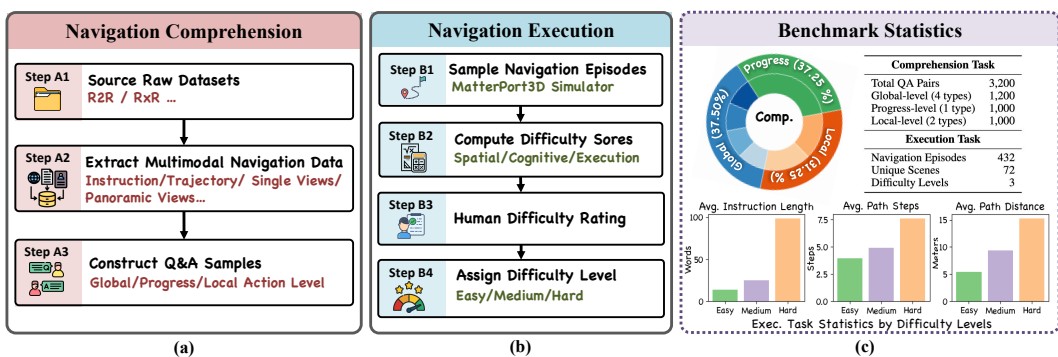

Figure 3: **NavBench construction pipeline and statistics.** (a) QA generation for comprehension tasks at global, progress, and local levels. (b) Execution pipeline combining automatic difficulty scoring and human ratings. (c) Benchmark statistics, including comprehension (comp.) task distribution, QA counts, and execution statistics (*e.g.*, instruction length, steps, distance).

simulator setup centers on abstracted decision-making, Section 4 illustrates how this framework can be extended to real-world navigation by converting viewpoint selection into low-level control.

Specifically, at each step, the model receives the current panoramic observation, the natural language instruction, and a list of candidate navigable viewpoints. The model must select the next location to move to, thereby executing the instruction step-by-step until the goal is reached. Formally, at each step $t$ of a navigation episode, the MLLM receives an instruction $\mathbf{x} = \{w_1, w_2, ..., w_L\}$ of length $L$, a set of candidate navigable observation viewpoints $\mathbf{O}_t = \{o_t^1, o_t^2, ..., o_t^N\}$, and optional context $\mathbf{C}_t$ (such as navigation history or previous actions). The agent must select an action $a_t$ corresponding to one of the navigable directions:

$$\mathbf{a}_t = \mathrm{MLLM}(\mathbf{x}, \mathbf{O}_t, \mathbf{C}_t). \tag{1}$$

This decision process may involve reasoning about the instruction, interpreting the current view, leveraging prior context, and anticipating the result of each candidate action.

## 3.2 Dataset Construction

**Data Sources** NavBench is constructed by reorganizing and enriching fine-grained navigation data with multimodal observations to enable zero-shot evaluation of MLLMs. We start by collecting instruction-trajectory pairs from multiple embodied navigation benchmarks, including R2R [9], RxR [30], GEL-R2R [56], and FGR2R [57]. These datasets serve as annotation sources, but do not include the visual inputs needed for multimodal reasoning. To address this gap, we use the Matterport3D simulator to extract both panoramic and single-viewpoint RGB images aligned with navigation trajectories. The image extraction process involves traversing agent paths, sampling intermediate viewpoints, and rendering corresponding visual observations. All visual and textual data are then organized into a unified structure that supports multiple reasoning tasks and enables consistent QA generation across comprehension and execution settings. Figure 3 shows the overall benchmark construction pipeline.

**Statistics** We report statistics in Figure 3(c), including distribution of comprehension subtasks and coverage of scenes and episodes in execution. These statistics reflect the scale and diversity of the benchmark across reasoning levels and scenes.

### 3.2.1 Question-and-Answer Pairs Collection

We design three diagnostic tasks targeting global alignment, temporal progress estimation, and local spatial and action reasoning. In total, we collect 3,200 question-and-answer pairs to evaluate comprehension capacity in embodied navigation.

**Global Instruction Alignment** To evaluate MLLMs' ability to align spatial trajectories with semantically consistent instructions, we construct a multiple-choice dataset comprising 1,200 examples. Each example consists of a panoramic trajectory and five candidate instructions, including one ground-truth

and four distractors. The distractors are generated using four perturbation strategies: (1) *Basic*: random instructions sampled from unrelated trajectories, testing global relevance; (2) *Directional replacements*, where spatial terms (*e.g.*, "left", "north") are substituted using POS tagging via NLTK, probing directional grounding; (3) *Object replacements*, where noun phrases are replaced with unrelated landmarks drawn from an external landmark-annotated dataset [56], evaluating object-trajectory grounding; (4) *Shuffled segments*, where human-annotated sub-instructions [57] are permuted to disrupt temporal structure while preserving grammaticality. Each instruction set is randomly ordered and paired with a panoramic trajectory composed of viewpoint sequences and movement annotations. The design promotes multimodal spatial reasoning and reduces reliance on superficial cues.

**Progress Estimation** This task is designed to evaluate a model's ability to perform temporal reasoning and monitor execution progress during navigation. Each full navigation instruction is segmented into a sequence of sub-instructions, and each sub-instruction is aligned with a corresponding portion of the agent's trajectory. We leverage fine-grained annotations [57], which provide this alignment between individual sub-instructions and the associated panoramic viewpoints traversed during execution. To construct evaluation examples, we truncate the trajectory at intermediate points that mark the end of specific sub-instructions. The model is presented with the truncated panoramic trajectory along with the full list of sub-instructions, and is required to predict the index of the last completed one. To ensure data quality and minimize ambiguity, we applied a combination of automatic filtering and manual validation to retain instruction-path pairs with well-defined temporal boundaries (details in Appendix). In total, we collect 1,000 such examples for evaluation.

**Local Observation-Action Reasoning** We design two multiple-choice reasoning tasks to evaluate a model's capacity for local spatial and action reasoning inference. Both tasks present ambiguous scenarios that require fine-grained visual discrimination and understanding of plausible transitions. In Future-Observation Prediction, the model receives a current view and an action, and must choose the correct resulting view from a set of candidates. In Future-Action Prediction, the model observes two consecutive views and selects the action that best explains the transition. For both tasks, distractors are carefully sampled from nearby observations or visually similar actions to ensure ambiguity and challenge. We collect 500 examples for each format, yielding a total of 1,000 samples. All questions are formatted as multiple-choice queries to ensure consistency across evaluation tasks.

### 3.2.2 Navigation Episodes Collection

We sample 432 navigation cases from 72 unique scenes in the Matterport3D simulator [55]. To systematically assess the difficulty of each case, we define a composite complexity score across three orthogonal dimensions: *spatial*, *cognitive*, and *execution* complexity. Each dimension is derived from structural properties of the environment or linguistic cues in the instruction, following the methodology inspired by [58, 59]. In addition, human evaluation is conducted to further support and validate the difficulty classification process.

**Spatial Complexity** It quantifies the geometric and topological challenges of a navigation trajectory. We consider four features: (1) total path length $d$, (2) standard deviation of turn angles $\theta$, (3) vertical range $z$ as a proxy for elevation change, and (4) 2D spatial area $A$ covered by the path. A binary indicator $\mathbb{I}(z > 1.5)$ is included to capture significant elevation changes such as floor transitions. These features are computed from agent poses and scene connectivity data. The spatial complexity score is defined as:

$$\Phi_{\text{spatial}} = \alpha_1 \cdot \log(1 + d) + \alpha_2 \cdot \log(1 + \theta) + \alpha_3 \cdot \mathbb{I}(z > 1.5) + \alpha_4 \cdot \log(1 + A). \tag{2}$$

**Cognitive Complexity** It reflects the linguistic difficulty of navigation instructions. We extract five features using dependency parsing: (1) instruction length $L$, (2) number of verbs $V$, (3) number of spatial terms $S$ (*e.g.*, *left*, *upstairs*), (4) number of landmark mentions $M$ (*e.g.*, *kitchen*), and (5) number of subordinate clauses $C$ (*e.g.*, `relcl`, `advcl`). The cognitive complexity score is defined as:

$$\Phi_{\text{cognitive}} = \beta_1 \cdot \log(1 + L) + \beta_2 \cdot \log(1 + V) + \beta_3 \cdot \log(1 + S) + \beta_4 \cdot \log(1 + M) + \beta_5 \cdot C. \tag{3}$$

**Execution Complexity** It measures the behavioral effort required to complete the navigation. We consider: (1) number of steps $N$, (2) number of turns $T$, (3) floor change indicator $F$, and (4) number of decision points $D$. The score is computed as:

$$\Phi_{\text{execution}} = \gamma_1 \cdot \log(1 + N) + \gamma_2 \cdot \log(1 + T) + \gamma_3 \cdot F + \gamma_4 \cdot D. \tag{4}$$

**Normalization** Each raw complexity score $\Phi$ is normalized to the range $[1, 9]$ using a non-linear mapping:

$$\hat{\Phi} = \text{round}\left(1 + 8 \cdot \frac{\log(1 + \Phi) - \log(1 + \Phi_{\min})}{\log(1 + \Phi_{\max}) - \log(1 + \Phi_{\min})}, 2\right). \tag{5}$$

The weights $\alpha$, $\beta$, and $\gamma$ are empirically set to balance the contribution of each factor.

**Human Evaluation** To complement the automatic scoring, we conducted a human evaluation to validate our difficulty annotations. A group of annotators independently rated each case along the three defined dimensions, using a 1–9 scale with detailed guidelines aligned to our scoring criteria. Further details are provided in the Appendix.

**Difficulty Categorization** Based on the final scores, each case is categorized into one of three levels, as illustrated in Figure 4:

- Easy (score 1–3): Short paths with simple instructions, few steps, minimal spatial reasoning, and clear landmarks.

- Medium (score 4–6): Instructions with moderate length, multiple landmarks or spatial phrases, and medium-length paths.

- Hard (score 7–9): Long trajectories guided by complex multi-step instructions, often involving floor transitions and multiple spatial references.

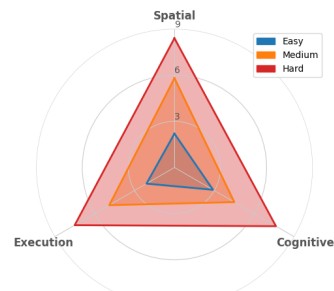

Figure 4: Radar chart of average complexity scores across cognitive, spatial, and execution dimensions for different difficulty levels.

## 4 Real-World Deployment Pipeline

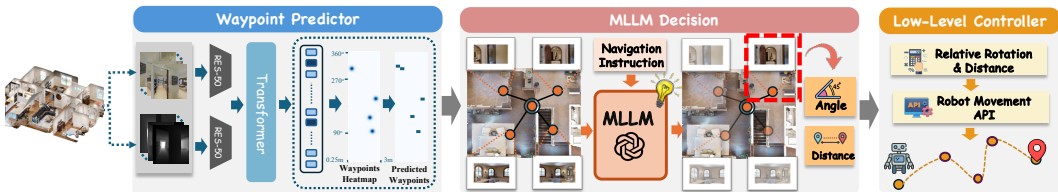

Figure 5: **Overview of the real-world embodied navigation pipeline.**

To demonstrate the real-world feasibility of MLLM-guided embodied navigation, we implement a modular pipeline that complements our benchmark evaluation, as illustrated in Figure 5. It consists of three modules: (1) a *Waypoint Predictor* that extracts RGB and depth inputs to generate candidate waypoints, (2) an *MLLM Decision Module* that selects the most goal-aligned waypoint, and (3) a *Low-Level Controller* that translates the selected waypoint into motion commands for execution on a physical robot. The system is deployed on a dual-arm mobile robot equipped with an RGB-D camera and evaluated in real indoor environments. More details are provided in the Appendix.

## 5 Evaluation on NavBench

### 5.1 Settings

**Models** We evaluate both proprietary and open-source MLLMs widely adopted in recent research. Proprietary models include GPT-4o, GPT-4o-mini, Gemini-2.0-flash. Open-source models include InternVL2.5-2B/8B [60], Qwen2.5-VL-3B/7B [61], LLaVA-OneVision-7B [62], LLaVA-Next-7B, and Llama3.2-Vision-11B [63].

**Implementation Details** Proprietary models are accessed via APIs, while open-source models are deployed using vLLM [64] and `lmdeploy` [65] on a single NVIDIA A6000 GPU (48GB). Simulator-based evaluations are conducted in the Matterport3D Simulator [55], built on high-resolution RGB-D scans of real indoor environments such as homes and offices. It provides realistic visual inputs and discrete agent movement within a 3D mesh, making it a standard testbed for embodied navigation. For real-world deployment, we integrate our pipeline with a dual-arm composite mobile robot equipped with an Intel RealSense D435 camera and a Water Drop 2 wheeled base. All physical experiments are conducted in a controlled indoor lab to assess robustness and feasibility.

Table 1: Performance comparison on **Navigation Comprehension** and **Execution**.

| Model | Navigation Comprehension | | | | Navigation Execution | | | | | | |
| | Global | Progress | Local | Comp. Avg | Easy | | Medium | | Hard | | Exec. Avg |
| | | Accuracy | | | SR | SPL | SR | SPL | SR | SPL | |
| Chance Level (Random) | 19.33 | 25.4 | 29.34 | 24.65 | 16.41 | 9.57 | 7.17 | 3.72 | 7.33 | 4.99 | 8.19 |
| *VLN-Bench (tiny) Performance* | | | | | | | | | | | |
| [†]Human Level | 88.33 | 79.00 | 85.00 | 84.11 | 91.67 | 88.68 | 87.50 | 81.53 | 75.00 | 65.17 | 81.59 |
| [†]GPT-4o | 51.67 | 45.00 | 63.00 | 53.89 | 66.08 | 49.01 | 43.79 | 36.44 | 25.00 | 20.11 | 40.07 |
| [†]Qwen2.5-VL-7B | 36.67 | 32.00 | 47.00 | 38.56 | 46.25 | 35.59 | 25.27 | 18.93 | 12.50 | 5.93 | 24.41 |
| *Closed Models* | | | | | | | | | | | |
| GPT-4o | 51.33 | 42.90 | **65.80** | 53.34 | **67.36** | **54.31** | 41.67 | 35.71 | **27.78** | **21.15** | **41.33** |
| GPT-4o-mini | 50.33 | 29.90 | 59.03 | 46.42 | 46.53 | 40.44 | 28.47 | 24.90 | 15.28 | 12.29 | 27.99 |
| Gemini-2.0-flash | **79.68** | 40.30 | 32.00 | 50.66 | 61.81 | 45.05 | **46.53** | **39.08** | 25.69 | 16.64 | 39.13 |
| o4-mini | 76.67 | **43.60** | 58.70 | **59.66** | 47.92 | 44.77 | 26.39 | 22.70 | 15.97 | 10.13 | 28.98 |
| *Open-Source Models* | | | | | | | | | | | |
| InternVL2.5-2B | **67.25** | 23.40 | 11.25 | 33.97 | 25.69 | 25.29 | 6.94 | 6.68 | 7.64 | 5.86 | 13.02 |
| Qwen2.5-VL-3B | 43.83 | 21.30 | **50.63** | 38.59 | 23.61 | 17.52 | 12.50 | 8.88 | 10.26 | 5.24 | 13.00 |
| InternVL2.5-8B | 62.75 | 28.50 | 28.12 | 39.79 | 28.47 | 28.19 | 7.66 | 7.42 | 7.64 | 6.18 | 14.26 |
| Qwen2.5-VL-7B | 57.58 | **31.20** | 47.00 | **45.26** | **41.67** | **32.55** | **22.92** | **17.43** | 10.42 | 5.67 | **21.77** |
| LLaVA-OneVision-7B | 31.17 | 26.60 | 39.00 | 32.26 | 31.25 | 17.64 | 15.58 | 7.80 | **15.02** | **7.84** | 15.86 |
| LLaVA-Next-7B | 38.33 | 27.40 | 28.50 | 31.41 | 27.08 | 25.95 | 11.81 | 7.69 | 7.64 | 6.07 | 14.54 |
| Llama3.2-Vision-11B | 36.00 | 23.40 | 29.10 | 14.75 | 27.08 | 25.90 | 10.42 | 9.19 | 10.02 | 7.60 | 15.04 |

*Note:* Dark teal and light teal indicate the top-performing closed and open-source models per column.
[†] indicates results evaluated on the NavBench (tiny) subset.

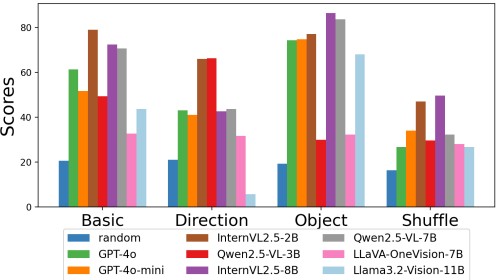

Figure 6: Model Performance under Different Instruction Perturbations.

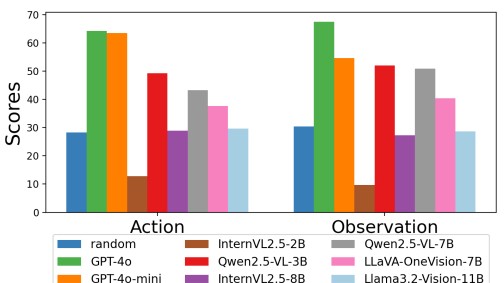

Figure 7: Model performance on Local Observation-Action Reasoning.

**Evaluation Metrics** Our benchmark includes both multiple-choice reasoning and embodied navigation execution tasks. For multiple-choice questions, we follow standard practice [5] and use *Accuracy* as the primary metric, which measures whether the model selects the correct answer from a set of candidates based on the provided information. For execution tasks, we adopt standard metrics in embodied navigation [9, 30]. *Success Rate (SR)* measures the percentage of episodes where the target object is visible from the agent's final viewpoint, defined as being within a 3-meter radius. *Success weighted by Path Length (SPL)* adjusts SR by path efficiency and is computed as:

$$\text{SPL} = \frac{1}{N} \sum_{i=1}^{N} S_i \cdot \frac{\ell_i}{\max(\ell_i, p_i)}. \qquad (6)$$

where $N$ is the number of episodes, $S_i \in \{0, 1\}$ indicates success, $\ell_i$ is the shortest path, and $p_i$ is the path length.

**VLN-Bench (tiny) Human Performance** To provide an upper-bound reference, we additionally report human performance on a compact subset of VLN-Bench, denoted as VLN-Bench (tiny), which was manually annotated and evaluated following the same protocol.

## 5.2 Performance

We begin by examining the relationship between comprehension and execution. As shown in Table 1, model performance on comprehension and execution tasks remains closely aligned. Among closed models, o4-mini achieves the highest comprehension average (59.66%) and maintains competitive execution performance (28.98%). GPT-4o follows with 53.34% and 41.33%, respectively, suggesting

Table 2: Impact of map information on GPT-4o.

| Diff. | Map | SR | SPL | Avg | Gain |
|-------|-----|-------|-------|-------|-------|
| Easy | ✗ | 67.36 | 54.31 | 60.84 | – |
|  | ✓ | 70.14 | 54.11 | 62.13 | +1.29 |
| Med. | ✗ | 41.67 | 35.71 | 38.69 | – |
|  | ✓ | 46.53 | 39.86 | 43.20 | +4.51 |
| Hard | ✗ | 27.78 | 21.15 | 24.47 | – |
|  | ✓ | 29.17 | 22.32 | 25.75 | +1.28 |

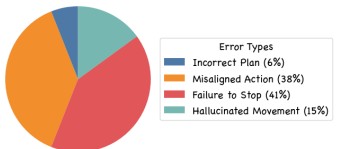

Figure 8: Distribution of navigation error types identified from manual analysis.

that o4-mini excels in understanding navigation instructions, while GPT-4o is stronger in executing them. Among open-source models, Qwen2.5-VL-7B achieves the best overall performance (45.26%, 21.77%), approaching GPT-4o-mini (46.42%, 27.99%) and demonstrating potential for practical deployment in real-world robotics. Turning to comprehension subtasks, InternVL2.5-2B performs strongly on Global Instruction Alignment (67.25%), even surpassing GPT-4o (51.33%). However, its accuracy drops sharply on more challenging reasoning tasks. In particular, Progress Estimation remains a consistent weakness across models; aside from GPT-4o (42.90%), all others perform poorly, highlighting current MLLMs' limitations in temporal reasoning. We next analyze how models perform across navigation difficulty levels. Most open-source models can only reliably complete Easy episodes, while GPT-4o maintains relatively strong results across all levels, suggesting better generalization. These findings suggest several overarching insights. First, comprehension and execution abilities are strongly linked. Second, temporal reasoning, particularly progress tracking, remains a major bottleneck. Third, compact open-source models like Qwen2.5-VL-7B can offer competitive performance with significantly lower resource requirements, making them promising for embodied applications.

## 5.3 Discussion

**Breakdown of Distractor Types in Instruction Alignment** We further analyze performance on the Global Instruction Alignment task by breaking down results across four distractor types: *basic*, *direction*, *object*, and *shuffle*. As shown in Figure 6, most models handle the basic condition well, indicating their ability to reject unrelated instructions. However, performance under *direction* and *object* perturbations varies significantly across models, suggesting inconsistent grounding of spatial terms and landmarks. Notably, all models perform poorly under the *shuffle* condition, where sub-instructions are reordered but their content remains unchanged. This result is particularly revealing: despite the presence of the same entities and actions, altering the temporal structure makes the instruction much harder for models to interpret. The models' failure in this setting highlights their limited ability to reason about temporal order within complex instructions. This finding aligns with the low scores observed in the Progress Estimation task, reinforcing that current MLLMs struggle with temporal understanding across both instruction-level and trajectory-level reasoning.

**Future-Action and Future-Observation Reasoning** We analyze performance on the Local Observation-Action Reasoning task, which includes two subtasks: Future-Action and Future-Observation Prediction. As shown in Figure 7, models show consistent performance across both, with GPT-4o clearly outperforming all others, consistent with its strong results in Navigation Execution. These subtasks reflect complementary reasoning skills. Future-Action Prediction tests whether a model can infer the spatial transition between two views, while Future-Observation Prediction requires anticipating how the environment changes after a given action. Both capabilities are critical for navigation, where agents should reason about cause and effect in spatial transitions.

**Effect of Map Information on Action Decisions** Although our benchmark evaluations assume no access to map information, reflecting real-world constraints, we investigate whether providing map connectivity can enhance action selection. Specifically, we follow the approach introduced in MapGPT, where topological relationships between explored nodes are encoded as text prompts. Using GPT-4o, we compare performance with and without map input across different difficulty levels. As shown in Table 2, the presence of map information consistently improves success rates, with the largest gain observed under medium difficulty, yielding an increase of 4.86 percentage points. This suggests that access to structured spatial context can facilitate better high-level reasoning and planning, especially in medium complexity settings where spatial ambiguity is more common.

Table 3: Performance comparison with and without CoT prompting.

| Model | Navigation Comprehension | | | | Navigation Execution | | | | | | |
| | Global | Progress | Local | Comp. Avg | Easy | | Medium | | Hard | | Exec. Avg |
| | Accuracy | | | | SR | SPL | SR | SPL | SR | SPL | |
| GPT-4o | 51.33 | 42.90 | 65.80 | 53.34 | 67.36 | 54.31 | 41.67 | 35.71 | 27.78 | 21.15 | 41.33 |
| GPT-4o + CoT | 60.42 | 40.20 | 60.75 | 53.79 | 61.11 | 49.04 | 44.44 | 36.88 | 30.56 | 23.20 | 40.87 |
| Qwen2.5-VL-7B | 57.58 | 31.20 | 47.00 | 45.26 | 41.67 | 32.55 | 22.92 | 17.43 | 10.42 | 5.67 | 21.77 |
| Qwen2.5-VL-7B + CoT | 60.50 | 31.40 | 48.00 | 46.63 | 43.75 | 33.06 | 22.92 | 15.62 | 11.11 | 7.19 | 22.28 |

**Effect of Chain-of-Thought** We incorporated Chain-of-Thought (CoT) prompting following [66] by prepending "Let's think step by step" to the instruction input. As shown in Table 3, experiments were conducted using two of the strongest models in our benchmark: GPT-4o and Qwen2.5-VL-7B. The results show that CoT prompting brings noticeable improvement in the Global Instruction Alignment task. GPT-4o improved by 9.09%, and Qwen2.5-VL-7B improved by 2.92%. However, the gains in other comprehension tasks were marginal or slightly negative. We hypothesize that simple CoT prompting does not sufficiently enhance performance in spatial or temporal reasoning tasks, which often require more structured, multi-step planning rather than generic step-by-step thinking. For the navigation execution task, we observed little benefit from CoT prompting. This is likely because the task itself already follows a step-by-step process: at each time step, the model receives the full instruction history and must decide the next action. Therefore, additional CoT prompting provides limited benefit in this context.

**Error Analysis** (1) We manually analyze 100 failed cases to understand model failures. Based on thought traces and action sequences, we identify four common error types: (a) *Incorrect Plan*: the plan misaligns with the instruction; (b) *Misaligned Action*: the plan is valid, but the chosen movement does not follow it; (c) *Failure to Stop*: the agent overshoots the goal or stops early; and (d) *Hallucinated Movement*: the model selects a nonexistent location. The error distribution is shown in Figure 8. These patterns align with weaknesses in comprehension tasks. For example, type (c) reflects poor *Progress Estimation*. This suggests execution failures often stem from temporal and spatial reasoning limitations, reinforcing the diagnostic value of NavBench.

(2) We further examine the impact of trajectory length on temporal reasoning. Test samples are grouped by length into short (1–2 steps), medium (3–4), and long (5+). For GPT-4o, the error rate increases from 35.3% (short) to 42.9% (medium) and 76.1% (long), showing that longer trajectories amplify temporal reasoning challenges. In contrast, weaker models such as LLaVA-OneVision-7B and InternVL2.5-2B maintain high error rates across all lengths, indicating persistent difficulty in progress estimation regardless of path complexity.

**Real-World Validation** To assess the feasibility of our real-world deployment pipeline, we conduct a pilot study in an indoor environment using GPT-4o and Qwen2.5-VL-7B, the top proprietary and open-source models from our benchmark. Each model is tested on 10 cases, achieving success rates of 60% and 40%, respectively. These results show that both can handle simple navigation tasks in real-world settings. Their success trends mirror execution performance in Table 1, where both models outperform others in their categories. This suggests that NavBench's simulation-based evaluation reliably reflects real-world embodied performance.

# 6   Conclusion

This paper presents NavBench, a diagnostic benchmark designed to evaluate MLLMs in embodied navigation under zero-shot settings. It decomposes the evaluation into two components: Navigation Comprehension, which evaluates global instruction alignment, temporal progress estimation, and local observation-action reasoning through three cognitively grounded tasks, and Navigation Execution, which examines step-by-step decision-making across varying levels of difficulty. Additionally, we develop a pipeline for real-world deployment of MLLM-driven agents. Through evaluation and targeted analysis, NavBench reveals limitations in temporal understanding and action grounding that are not captured by standard success metrics. It also shows that lightweight open-source models can be effective in simpler navigation scenarios. We hope NavBench can serve as a useful resource for analyzing the embodied capabilities of MLLMs and supporting future work in this direction.

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
