# OpenReview forum: "NavBench: Probing Multimodal Large Language Models for Embodied Navigation"
_NeurIPS.cc/2025/Conference — NeurIPS 2025 poster_

### Official Review · Reviewer_eAcf · 2025-06-21

**Clarity:** 3
**Significance:** 2
**Originality:** 2
**Rating:** 4
**Confidence:** 4

**Summary:**

This paper proposes a benchmark with Multimodal Large Language Models for Embodied Navigation. Two components are included in this work, navigation comprehension and execution. This work also supports that converts MLLMs' outputs into robotic actions.

**Questions:**

- The main concerns are listed in the Weaknesses section.

**Ethical Concerns:**

["NO or VERY MINOR ethics concerns only"]

**Final Justification:**

The author provides more details about experiments and motivation. I recognize their efforts about exploring MLLMs in embodied navigation, and give a positive final score.

**Limitations:**

Yes

**Paper Formatting Concerns:**

The formatting of this paper requires further refinement. Here are several specific suggestions:(1) All equations should be followed by appropriate punctuation (commas or periods); (2) For bullet points, the first letter of each sub-point should be capitalized; (3) Abbreviations such as *e.g.* should be italicized; (4) The citation style of references should be consistent throughout the manuscript; (5) All figures and tables should be presented as vector graphics (currently, Figures 4 and 8 are raster images); (6) Formulas in the Metrics section should be displayed on a separate line; (7) Vectors and matrices in equations should be formatted in bold italics; (8) In typesetting, each line should contain more than one third of its maximum possible word count to enhance readability.

**Quality:**

2

**Strengths And Weaknesses:**

- Strengths
    - This paper is will written, and easy to follow.
    - This paper proposes a navigation benchmark specifically for MLLMs, focusing on exploring their long-term processing capabilities in 3D scene.
    - This paper implements a systematic comparison and analysis of the performance trends of closed-source and open-source MLLMs in embodied reasoning and decision-making.

- Weakness
    - This paper proposes a new benchmark, which is mainly constructed based on existing datasets such as R2R and RxR, and the evaluation metrics lack novelty. I recommend that the authors further consider designing new input-output formats and evaluation criteria to comprehensively assess the spatial reasoning abilities of multimodal large models.
    - This paper proposes a pipeline for real-world deployment, but the current description is rather general. I suggest the authors provide additional details regarding the generation process of the waypoint heatmap, the algorithm used to extract predicted waypoints, and the specific methods for calculating the angles and distances between different waypoints. Including algorithmic descriptions and visualizations of these steps would significantly improve the clarity and reproducibility of the pipeline.
    - This paper evaluates proprietary MLLMs (e.g., GPT-4o) and open-source MLLMs (e.g., InternVL2.5-2B). I have the following questions and suggestions: (1) Have the authors considered including more open-source models (e.g., Gemini series) and proprietary models (e.g., LLaVA-Next) to enrich the comparison? (2) In Table 1, the open-source models cover a wide range of parameter sizes (e.g., 2B, 3B, 7B, 11B). For fair comparison, it is recommended to match model sizes as much as possible; (3) It would be beneficial to see more ablation studies to provide a finer-grained analysis of the current MLLMs’ performance and limitations in zero-shot navigation tasks within 3D environments.

---

> ### Author Rebuttal · Authors · 2025-07-30
>
> We thank the reviewer for taking the time to provide detailed feedback. We have carefully reviewed all the comments and provide point-by-point responses below. We hope these clarifications help resolve any potential misunderstandings and more clearly convey the contributions and intentions of our work.
>
> ---
>
> > **Q1:** Novelty concern regarding reliance on existing datasets and suggestion to design new input/output formats and evaluation metrics.
>
> **A1:** We understand the reviewer’s concerns, which may reflect a possible misunderstanding about the novelty of our task design and evaluation setup. We clarify each point below:
>
> **(1) Use of existing datasets.**
> Our benchmark utilizes navigation paths and instructions from R2R and RxR due to their wide adoption and strong coverage of realistic indoor scenes. However, we did not simply concatenate or reuse these datasets. All visual inputs were re-collected using the Matterport3D simulator to align with our specific task requirements, and the underlying data were reorganized to support our newly designed tasks.
>
> Repurposing established datasets is a widely accepted practice in benchmark design, as demonstrated by recent works such as VSI-Bench [1] and EmbodiedScan [2]. What distinguishes our work is the construction of entirely new evaluation tasks, including three comprehension tasks (global alignment, temporal progress estimation, and local action reasoning) and an execution task. These are based on restructured inputs and newly defined outputs, supporting a zero-shot evaluation protocol for MLLMs that is fundamentally different from prior trajectory-following settings.
>
> **(2) Evaluation metrics.**
> We adopt QA-style accuracy for comprehension and use step-wise success rate along with standard navigation metrics such as SR and SPL for execution. Although these metrics are commonly used in prior work, we found them to be practical and interpretable in the context of our newly designed tasks. They allow us to capture model performance at different stages of navigation, from high-level understanding to low-level execution.
>
> **(3) Designing new input/output formats and assessing spatial reasoning.**
> Thank you for the suggestion. While our benchmark does not isolate spatial reasoning as a standalone capability, it is explicitly embedded in tasks such as Local Observation-Action Reasoning. In this task, the model is required to predict how the visual scene would change after taking a specific action, which involves understanding the spatial layout and action affordances within the 3D environment. In future extensions of the benchmark, we will consider designing additional spatial reasoning tasks tailored to embodied navigation scenarios.
>
> ---
> > **Q2:** More details of the real-world deployment pipeline.
>
> **A2:** Thank you for your suggestion. As we were constrained by the main paper’s space limit, we included the detailed description of the deployment pipeline in the supplementary material (Section C). This includes panoramic image construction, depth estimation, feature fusion with dual ResNet-50 backbones, transformer-based heatmap generation, waypoint selection using NMS, and translation into robot actions (Equations 1–2).
>
> While additional visualizations cannot be included here due to formatting limits, we will incorporate them in the revised version to improve clarity. These materials will also be released alongside our code to support reproducibility.
>
> ---
> > **Q3:** Suggestions to (1) include more models (e.g., Gemini, LLaVA-Next) for broader comparison, (2) control for model size across comparisons, and (3) conduct more ablation studies.
>
> **A3:** Thank you for the valuable suggestions. We address each point below:
>
> **(1) Inclusion of more models.**
> We evaluated additional models to further enrich the comparison as suggested. This includes both proprietary models (e.g., Gemini-2.0-flash, OpenAI o4-mini) and open-source models (e.g., LLaVA-Next-7B). Results are summarized below:
>
> **Table 1: Performance comparison on Navigation Comprehension and Execution**
> | Model               | Global | Progress | Local  | Comp. Avg | Easy SR | Easy SPL | Med. SR | Med. SPL | Hard SR | Hard SPL | Exec. Avg |
> |---------------------|--------|----------|--------|-----------|---------|----------|--------|----------|---------|----------|-----------|
> | **Gemini-2.0-flash**           | 79.68  | 40.30    | 32.00  | 50.66     | 61.81   | 45.05    | 46.53  | 39.08    | 25.69   | 16.64    | 39.13     |
> | **o4-mini**   | 76.67  | 43.60    | 58.70  | 59.66     | 47.92   | 44.77    | 26.39  | 22.70    | 15.97   | 10.13     | 28.98     |
> | **LLaVA-Next-7B**    | 38.33  | 27.40    | 28.50  | 31.41     | 27.08   | 25.95    | 11.81  | 7.69    | 7.64   | 6.07     | 14.54     |
>
> *Note: Comp. = Comprehension; Exec. = Execution; SR = Success Rate; SPL = Success weighted by Path Length; Med. = Medium.*
>
> **(2) Controlling for model size.**
> Since Open-source models vary considerably in size and architecture, it is a bit challenging to make them exact size matching. Thus, we grouped models into approximate size tiers (e.g., 2–3B, 7–8B) to ensure general comparability while covering a diverse set of representative models. We will continue to expand the benchmark with additional models organized by size to facilitate more fine-grained comparisons.
>
> **(3) Additional ablation studies.**
> We conducted an ablation study to assess the effect of Chain-of-Thought (CoT) prompting by prepending “Let’s think step by step” to the text input, following [3]. For a representative comparison, we selected one proprietary model (GPT-4o) and one open-source model (Qwen2.5-VL-7B) based on their strong overall performance.
>
> **Table 2: Performance Comparison with and without CoT Prompting**
> | Model               | Global | Progress | Local  | Comp. Avg | Easy SR | Easy SPL | Med. SR | Med. SPL | Hard SR | Hard SPL | Exec. Avg |
> |---------------------|--------|----------|--------|-----------|---------|----------|--------|----------|---------|----------|-----------|
> | **GPT-4o**           | 51.33  | 42.90    | 65.80  | 53.34     | 67.36   | 54.31    | 41.67  | 35.71    | 27.78   | 21.15    | 41.33     |
> | **GPT-4o + CoT**     | 60.42  | 40.20    | 60.75  | 53.79     | 61.11   | 49.04    | 44.44  | 36.88    | 30.56   | 23.20    | 40.87     |
> | **Qwen2.5-VL-7B**    | 57.58  | 31.20    | 47.00  | 45.26     | 41.67   | 32.55    | 22.92  | 17.43    | 10.42   | 5.67     | 21.77     |
> | **Qwen2.5-VL-7B + CoT** | 60.50  | 31.40    | 48.00  | 46.63     | 43.75   | 33.06    | 22.92  | 15.62    | 11.11   | 7.19     | 22.28     |
>
> *Note: Comp. = Comprehension; Exec. = Execution; SR = Success Rate; SPL = Success weighted by Path Length; Med. = Medium.*
>
> We observed that CoT prompting yielded moderate gains in global instruction alignment but had limited or even negative effects in spatial and temporal reasoning tasks. For the execution task, CoT brought little benefit, likely because our step-wise formulation already structures the decision-making process.
>
> We will include these results and tables in the revised version.
>
> ---
> > **Q4:** The formatting of this paper requires further refinement.
>
> **A4:** Thank you for the suggestions. We will carefully revise the manuscript to address the listed formatting issues and these improvements will be reflected in the revised version.
>
> ---
> We sincerely hope our responses help clear up any confusion and alleviate your concerns.
>
> ---
> ### References
> [1] *Thinking in Space: How Multimodal Large Language Models See, Remember, and Recall Spaces*. Proceedings of the IEEE/CVF Conference on Computer Vision and Pattern Recognition (CVPR), 2025.
> [2] *EmbodiedScan: A Holistic Multi-Modal 3D Perception Suite Towards Embodied AI*. Proceedings of the IEEE/CVF Conference on Computer Vision and Pattern Recognition (CVPR), 2024.
> [3] *Chain-of-thought prompting elicits reasoning in large language models*. Advances in Neural Information Processing Systems (NeurIPS), 2022.

---

> > ### Comment · Reviewer_eAcf · 2025-08-02
> >
> > After reading the author's rebuttal, I think the motivation of this paper is meaningful. The author also provides more results about other MLLMs. These results should be included in the updated version. I will raise my score.

---

> > > ### Author Response · Authors · 2025-08-02
> > > **Thanks to Reviewer eAcf**
> > >
> > > We sincerely appreciate your thoughtful feedback and your willingness to reconsider the score. Your recognition of our motivation and the additional results is very encouraging. We will incorporate the suggested revisions into the final version. Thank you again for your support.

---

### Official Review · Reviewer_5z5E · 2025-07-03

**Clarity:** 4
**Significance:** 3
**Originality:** 2
**Rating:** 5
**Confidence:** 3

**Summary:**

This paper presents NavBench, a benchmark designed to evaluate the embodied navigation capabilities of Multimodal Large Language Models (MLLMs) under zero-shot settings.

The benchmark consists of two components: (1) navigation comprehension tasks evaluating global instruction alignment, temporal progress estimation, and local observation-action reasoning; (2) step-by-step execution tasks across different difficulty levels.

The authors evaluate both proprietary and open-source MLLMs on NavBench, highlighting persistent challenges in temporal reasoning.

**Questions:**

1. Could the authors provide more details about the quality control methods of constructed Temporal Progress Estimation data?

2. Regarding the complexity score weights (α,β,γ), the paper states they are "empirically set."  Was there a specific calibration step against the human ratings, or were they chosen based on prior work and intuition?

**Ethical Concerns:**

["NO or VERY MINOR ethics concerns only"]

**Final Justification:**

The paper quality is good, and the authors have addressed my concerns. I will keep my positive rating.

**Limitations:**

The paper would benefit from a more detailed discussion of potential biases in the constructed QA pairs.

**Quality:**

3

**Strengths And Weaknesses:**

## Strengths
1. The paper addresses a clear gap in evaluating MLLMs for embodied navigation, moving beyond static vision-language tasks to dynamic, sequential decision-making scenarios.

2. The three-level comprehension assessment (global, progress, local) is cognitively grounded and provides diagnostic insights.

3. The systematic categorization of tasks by spatial, cognitive, and execution complexity with both automatic scoring and human validation is methodologically sound.

4. The paper evaluates multiple models across different scales and provides a detailed analysis of failure modes, revealing important insights about temporal reasoning limitations.

## Weaknesses
1. For Temporal Progress QA pairs construction, how the authors do the filtering ("using a curated list of valid instruction-path pairs") is not clear.

2. The weights α, β, γ for difficulty scoring are "empirically set" but the justification is unclear.

---

> ### Author Rebuttal · Authors · 2025-07-30
>
> We sincerely thank the reviewer for the constructive and encouraging feedback. We provide clarifications below for the points raised.
>
> ---
> > **Q1:** More details about the quality control methods of constructed Temporal Progress Estimation data.
>
> **A1:** Thank you for raising this point. The QA pairs for the Temporal Progress Estimation task were selected from a larger candidate pool through a combination of automatic filtering and manual review.
>
> We first applied an automatic rule to exclude QA pairs in which two adjacent sub-instructions ended at the same viewpoint, as such cases typically indicate unclear temporal boundaries. After removing these structurally ambiguous pairs, we manually reviewed the remaining candidates and excluded samples that lacked a clear or meaningful sense of temporal progress, such as those where the chunk images showed minimal spatial change or the sub-instruction-to-trajectory alignment was unclear.
>
> This two-stage filtering process ensured that the final QA pairs exhibit visually and semantically distinct progression steps. We will include these filtering details in the revised version.
>
> ---
> > **Q2:** Clarification on complexity score weights.
>
> **A2:** Thanks for your question. To determine the weights in our complexity scoring framework, we conducted a detailed statistical analysis of the numerical features associated with each complexity dimension: spatial, cognitive, and execution.
>
> First, we extracted structured features from each trajectory. For spatial complexity, we included path length, standard deviation of turning angles, elevation range, and two-dimensional area coverage. Cognitive complexity was based on instruction length, number of verbs, number of spatial terms, number of landmark mentions, and number of subordinate clauses. Execution complexity was measured by the number of steps, number of turns, presence of floor changes, and number of decision points.
>
> Second, we sampled data from 50 scenes and computed summary statistics for each feature, including mean, standard deviation, and min–max range. This analysis helped us understand the scale and variation of each feature. For example, instruction length exhibited a broader numerical spread than the number of landmark mentions, which would otherwise lead to a disproportionate impact on the cognitive complexity score.
>
> Third, we initialized all feature weights equally within each complexity category and then adjusted them based on the observed distributions to ensure balanced contributions. Features with inherently larger value ranges were assigned smaller weights to avoid dominating the overall score. We also tuned the aggregation weights across the three complexity dimensions to ensure a balanced contribution to the final difficulty score.
>
> During the adjustment process, we also considered the distribution of human difficulty ratings to improve the alignment between automatic scores and perceived complexity. To evaluate the robustness of the weighting scheme, we performed a sensitivity analysis by perturbing each weight by 20 percent while keeping the others fixed. As shown in Figure 1 (supplementary), over 90 percent of samples retained their original difficulty category, indicating that the scoring method is relatively stable.
>
> These details will be incorporated into the revised manuscript to improve clarity.

---

### Official Review · Reviewer_uQaG · 2025-07-07

**Clarity:** 4
**Significance:** 3
**Originality:** 3
**Rating:** 5
**Confidence:** 4

**Summary:**

This paper introduces an evaluation benchmark for multimodal language models for vision-and-language embodied navigation tasks, specifically in zero-shot settings. Their methodology proposes evaluating both the comprehension as well as execution in the navigation task. They also develop a pipeline for real-world deployment of these agents in environments. They show how this analysis can reveal limitations in current models in temporal understanding and other factors, that previous metrics obfuscate; and also show how current zero-shot models can be thoroughly tested for such environments.

**Questions:**

1. There are now a few vision and language datasets that come in multiple languages (e.g, english -> trajectories, hindi -> trajectories). It would be worth discussing whether this evaluation, especially for the comprehension component, needs to factor a language id in, and if there would be any differences expected. If not (which, OTTH, there should not be currently) it would be worth pointing out that a strength of this methodology is that it is language agnostic, and also, future work on the multilingual datasets might be interesting insights from such metrics.
2. Can we get an upperbound on model performance for VLN models? The numbers in the table are fairly low, which is expected for zero-shot LLMs, but the best-performing VLN models (that are trained specifically for this task) would be good to see alongside, to calibrate the difference.

**Ethical Concerns:**

["NO or VERY MINOR ethics concerns only"]

**Limitations:**

Yes

**Quality:**

3

**Strengths And Weaknesses:**

1. The evaluation proposed by this paper is important to the field of vision/language navigation and mulltimodal models.
2. It draws from relevant existing work to use metrics that are well-tested and thought out (e.g., not just goal accuracy and trajectory metrics, but weighted averages of them and so forth) and combines them into one benchmark,
3. They show how this benchmark reveals weaknesses of models that previous benchmarks and metrics could not easily delineate.
4. It is well written and explained and all metrics are easily reproducible and will be useful for future work.
5. They illustrate how models they evaluate can be deployed in the real world, specifically on a dual-arm robot.
6. They evaluate a sufficient range of models (including the state-of-the-art LLMS (although they might not be the best VLN models).

---

> ### Author Rebuttal · Authors · 2025-07-30
>
> We greatly appreciate your positive and constructive feedback. Thank you for recognizing the value of our work. We provide clarifications on your insightful suggestions below.
>
> ---
> > **Q1:** Multilingual support in the benchmark.
>
> **A1:**  Thank you for the constructive suggestion regarding multilingual evaluation. While our current benchmark is conducted in English, we agree that assessing model generalization across languages is a valuable direction.
>
> To explore this, we conducted a preliminary study on the Global Instruction Alignment task, which is part of our navigation comprehension evaluation. This task involves aligning a language instruction with a visual trajectory and is particularly suitable for analyzing the effect of instruction language, as it directly tests the model’s ability to semantically match textual descriptions with visual paths.
>
> We selected two representative models, GPT-4o (the strongest closed-source model) and Qwen2.5-VL-7B (the best-performing open-source model), and translated 30 randomly sampled instructions into four languages. Two of them, Italian and German, are typologically close to English, while the other two, Hindi and Telugu, are linguistically distant and included in the multilingual RxR dataset. To isolate the effect of instruction language, the rest of the prompt was kept in English. The translations were performed automatically without human post-editing to simulate a zero-shot cross-lingual setting.
>
> The accuracy results  are shown below:
>
> **Table 1: Multilingual Global Instruction Alignment Accuracy**
>
> | Model              | English | Hindi | Telugu | Italian | German |
> |--------------------|---------|-------|--------|---------|--------|
> | GPT-4o             | 55.00   | 37.50 | 49.17  | 45.00   | 51.67  |
> | Qwen2.5-VL-7B      | 62.50   | 48.33 | 32.50  | 60.83   | 54.17  |
>
> We observe that performance degradation is relatively minor for Italian and German, while both models exhibit noticeable drops on Hindi and Telugu. This supports the reviewer’s intuition that language differences can impact model performance, particularly when the language diverges significantly from English.
>
> Since this is a small-scale pilot study based on automatic translation without human post-editing, we treat the results as exploratory. In future work, we plan to expand multilingual evaluation with larger-scale experiments and explore the inclusion of language ID tokens to improve cross-lingual robustness. We will summarize this discussion in the revised version.
>
> ---
> > **Q2:** Upperbound performance.
>
> **A2:** Thank you for this suggestion. Including an upper bound is indeed important for interpreting zero-shot MLLM performance. Due to time constraints, we are currently unable to provide results from task-specific trained VLN models, but we plan to include them in future versions for a more complete comparison.
>
> In the meantime, we provide a human performance reference derived from a compact subset of the benchmark. Given the scale of the dataset, full human annotation is infeasible, so we randomly sampled 392 representative instances across all tasks. The resulting accuracy (84.11% for comprehension and 81.59% for execution) serves as a practical empirical upper bound for interpreting current MLLM performance.
>
> ---
> We thank the reviewer again for the thoughtful suggestions. We believe these additions will further improve the clarity and scope of our work, and we will incorporate the relevant discussions into the revised version.

---

### Official Review · Reviewer_vvYt · 2025-07-11

**Clarity:** 3
**Significance:** 2
**Originality:** 2
**Rating:** 5
**Confidence:** 4

**Summary:**

This work introduces NavBench to evaluate the multimodal large language models' capabilities, including navigation comprehension and navigation execution capabilities in VLN. Experiments and analyses with 2 closed models and 4 open-sourced models on the NavBench benchmark show some interesting findings.

**Questions:**

- The length of trajectories matters for temporal progress estimation. So is it possible to do an analysis about the relationship between errors and trajectory length?
- The paper does not mention whether chain-of-thought thinking is used for models' predictions. Considering there are some discussions about whether thinking is useful for navigation tasks in recent literature, I suggest to add some comparisons between using thinking and without thinking.
- Considering the reasoning capability of thinking models, is it possible to test performance of some reasoning models such as o4-mini in this benchmark?

**Ethical Concerns:**

["Major Concern: Data quality and representativeness"]

**Final Justification:**

All my concerns proposed in my weakness and questions, except the use of panoramic images instead of egocentric images, are addressed by the authors' responses. So I would like to raise my score from 4 to 5.

**Limitations:**

yes

**Paper Formatting Concerns:**

No.

**Quality:**

3

**Strengths And Weaknesses:**

- Strengths:
	- The paper is well-written and easy to follow.
	- The paper introduces three navigation comprehension tasks and one navigation execution task to evaluate the commonly used VLMs' capabilities. The introduced tasks are carefully designed and expected to be quite useful to evaluate thee VLMs' capabilities in the context of navigation.
	- The tested models are quite diverse, including both proprietary and open-source models.
	- The proposed difficulty level for navigation execution tasks is novel and useful.
- Weaknesses:
	- The paper is a benchmark paper but not submitted to the benchmark track. So we cannot really check the content and quality of the benchmark. Moreover, the paper does not discuss the license regarding the benchmark release. Considering that Matterport3D is not fully open-sourced and has its own license, so it is important to discuss the license if the authors plan to release the benchmark.
	- The paper uses panoramic images as observation format for all the tasks in the benchmark. This is no longer the common setting for the VLN-CE task, which is a more popular task nowadays. The VLN-CE task usually adopts egocentric videos as observation input, so the findings of this paper may have limit to extend to it.
	- The navigation execution task in the benchmark, where a model is required to pick up the next-step viewpoint, is almost the same as the common setting of VLN task. So there is a concern why not directly use the standard VLN task?

---

> ### Author Rebuttal · Authors · 2025-07-30
>
> We appreciate your constructive feedback and suggestions. Please find our responses to each comment below.
>
> ---
> > **Q1:** Benchmark transparency and licensing concerns.
>
> **A1:** Thank you for raising these concerns. We agree that benchmark transparency and reproducibility are essential. To support this, we provide detailed descriptions in Section 3 of the main paper and include additional details, sample annotations, and evaluation scripts in the supplementary material. We hope this allows reviewers to assess the quality and reproducibility of the benchmark.
>
> Regarding licensing, we strictly adhere to Matterport3D’s academic usage terms. We do not release any rendered images or meshes. Instead, we provide structured annotations (e.g., instructions, trajectories, and QA pairs) and scripts that allow users to reproduce the visual input via the official simulator. This practice follows the precedent set by benchmarks like R2R. A licensing statement will be included in the final release to ensure compliance.
>
> ---
> > **Q2:** Use of panoramic images instead of egocentric video in VLN-CE tasks.
>
> **A2:** Although egocentric videos have become a common input format in recent VLN-CE tasks, panoramic observations remain a viable and widely adopted setting in prior and recent works. For example, both ETPNav [1] and Open-Nav [2] conduct experiments on VLN-CE with panoramic observations, highlighting their utility in evaluating high-level navigation reasoning.
>
> We choose panoramic inputs for two key reasons. First, panoramic observations offer a complete 360-degree view of the environment, which provides better spatial grounding for language-conditioned navigation and reduces the ambiguity that may arise from a narrow field-of-view. This is particularly important when evaluating multimodal large language models (MLLMs), which benefit from a richer global context to perform reasoning and instruction alignment.
>
> Second, panoramic views can be practically implemented in real-world navigation systems. In our real-robot deployment, we rotate the robot in place to capture 12 directional images, forming a panoramic observation. This setup shows that panoramic input is not limited to simulation and can be readily applied in real environments.
>
> Therefore, although our benchmark currently adopts panoramic views, we consider it compatible with the goals of VLN-CE evaluation and can provide complementary insights into model capabilities. We also consider extending our benchmark to support egocentric video inputs in future iterations.
>
> ---
> > **Q3:** Why not directly use the standard VLN task for execution?
>
> **A3:** While our execution task shares the same basic formulation as the standard VLN setting, where the agent selects the next-step viewpoint, our design emphasizes a different evaluation focus. We aim to assess the capabilities of MLLMs under zero-shot conditions, without any task-specific training or fine-tuning.
>
> A key distinction lies in our explicit difficulty-aware design. Instead of treating all trajectories uniformly, we define a composite complexity score across three orthogonal dimensions: spatial, cognitive, and execution complexity. This allows us to categorize the tasks into three difficulty levels (easy, medium, and hard) and conduct fine-grained evaluation. For example, GPT-4o achieves a success rate of 67.36% on the easy subset, suggesting that it may already be suitable for deployment in relatively simple real-world scenarios.
>
> ---
> > **Q4:** Error analysis on trajectory length in progress estimation.
>
> **A4:** We appreciate this insightful suggestion. We agree that trajectory length may significantly affect a model’s ability to estimate temporal progress. To investigate this, we grouped the test samples by trajectory length into three buckets: short (1–2 steps), medium (3–4), and long (5 or more), and computed the error rates for different models. For GPT 4o, the strongest overall performer, we observed a clear increase in error rate from 35.3% (short) to 42.9% (medium) and 76.1% (long). This indicates that longer trajectories pose greater challenges for temporal reasoning, possibly due to accumulated ambiguity or compounding errors. In contrast, lower-performing models such as LLaVA-OneVision-7B and InternVL2.5-2B show consistently high error rates across all lengths (often above 80–90%), suggesting they struggle regardless of trajectory complexity and may operate close to random guessing. We will include this analysis in the revised version.
>
> ---
> > **Q5:** Chain-of-thought prompting for navigation tasks.
>
> **A5:** Thank you for the helpful suggestion. We incorporated Chain of Thought (CoT) prompting following [3] by prepending “Let’s think step by step” to the instruction input. We conducted experiments using two of the strongest models in our benchmark: GPT-4o and Qwen2.5-VL-7B.
>
> Our results show that CoT brings noticeable improvement in the Global Instruction Alignment task. GPT-4o improved by 9.09 percent, and Qwen2.5-VL-7B improved by 2.92 percent. However, the gains in other comprehension tasks were marginal or slightly negative. We hypothesize that simple CoT prompting may not sufficiently enhance the model’s performance in spatial and temporal reasoning tasks, which often require more structured or multi-step planning rather than generic step-by-step thinking.
>
> For the navigation execution task, we observed little benefit from CoT prompting. This is likely because the task is already structured in a step-by-step fashion. At each step, the model receives the full instruction history and must decide on the next action. Therefore, additional prompting provides limited benefit.
>
> We will include this analysis and the following performance comparison table in the revised version:
>
> **Table 1: Performance Comparison with and without CoT Prompting**
>
> | Model               | Global | Progress | Local  | Comp. Avg | Easy SR | Easy SPL | Med. SR | Med. SPL | Hard SR | Hard SPL | Exec. Avg |
> |---------------------|--------|----------|--------|-----------|---------|----------|--------|----------|---------|----------|-----------|
> | **GPT-4o**           | 51.33  | 42.90    | 65.80  | 53.34     | 67.36   | 54.31    | 41.67  | 35.71    | 27.78   | 21.15    | 41.33     |
> | **GPT-4o + CoT**     | 60.42  | 40.20    | 60.75  | 53.79     | 61.11   | 49.04    | 44.44  | 36.88    | 30.56   | 23.20    | 40.87     |
> | **Qwen2.5-VL-7B**    | 57.58  | 31.20    | 47.00  | 45.26     | 41.67   | 32.55    | 22.92  | 17.43    | 10.42   | 5.67     | 21.77     |
> | **Qwen2.5-VL-7B + CoT** | 60.50  | 31.40    | 48.00  | 46.63     | 43.75   | 33.06    | 22.92  | 15.62    | 11.11   | 7.19     | 22.28     |
>
> *Note: Comp. = Comprehension; Exec. = Execution; SR = Success Rate; SPL = Success weighted by Path Length; Med. = Medium.*
>
> ---
> > **Q6:** Performance of reasoning-focused models such as o4-mini.
>
> **A6:** Based on your valuable suggestion, we included the reasoning-focused model o4-mini in our evaluation. As shown in the table below, o4-mini demonstrates strong performance on the comprehension tasks, particularly on the Global Instruction Alignment task. It achieves the highest average comprehension score among all models, outperforming GPT-4o’s 53.34.
>
> However, its performance on the navigation execution task is less impressive and is generally on par with GPT-4o-mini. This aligns with our observation from the CoT experiments: o4-mini appears to be better suited for reasoning-based tasks that involve question answering and language understanding, but shows limitations when applied to decision-making tasks that require grounding and multi-step action planning.
>
> These findings and discussions will be incorporated into the revised version.
>
> **Table 2: Performance of o4-mini across comprehension and execution tasks.**
>
> | Model     | Global | Progress | Local  | Comp. Avg | Easy SR | Easy SPL | Med SR | Med SPL | Hard SR | Hard SPL | Exec. Avg |
> |-----------|--------|----------|--------|-----------|---------|----------|--------|----------|---------|----------|-----------|
> | o4-mini   | 76.67  | 43.60    | 58.70  | 59.66     | 47.92   | 44.77    | 26.39  | 22.70    | 15.97   | 10.13     | 28.98     |
>
> *Note: Comp. = Comprehension; Exec. = Execution; SR = Success Rate; SPL = Success weighted by Path Length; Med. = Medium.*
>
> ---
> We hope these clarifications and additional experiments address your concerns. Thank you again for your encouraging feedback and constructive suggestions.
>
> ---
> ### References
> [1] *ETPNav: Evolving Topological Planning for Vision-Language Navigation in Continuous Environments*. IEEE Transactions on Pattern Analysis and Machine Intelligence (TPAMI), 2024.
> [2] *Open-Nav: Exploring Zero-Shot Vision-and-Language Navigation in Continuous Environment with Open-Source LLMs*. Proceedings of the IEEE International Conference on Robotics and Automation (ICRA), 2025.
> [3] *Chain-of-thought prompting elicits reasoning in large language models*. Advances in Neural Information Processing Systems (NeurIPS), 2022.

---

> > ### Comment · Reviewer_vvYt · 2025-08-06
> >
> > Thanks for the authors' responses. I have read all the reviews and corresponding responses. They have addressed most of my concerns for this paper. I would like to raise my score from 4 to 5.

---

> ### Author Response · Authors · 2025-08-06
> **Thanks to Reviewer vvYt**
>
> Thank you sincerely for carefully reviewing our rebuttal and providing such encouraging feedback. We are truly grateful for your thoughtful comments and constructive suggestions. We will carefully incorporate the relevant discussions from the rebuttal into the final version of the paper to reflect your valuable suggestions. Your support and insights have been invaluable in improving the quality of our work. Thank you again for your time and thoughtful engagement with our work.

---

### Note · Authors · 2025-08-12

We sincerely thank the Area Chair and all reviewers for the time and effort they dedicated to evaluating our work and engaging in the discussion. We appreciate that the reviewers recognized the clear motivation behind this work, the care taken in designing the benchmark, and the practical contributions it offers to evaluating MLLMs for embodied navigation, as well as the thoughtful suggestions, which provide valuable directions to further extend the benchmark’s scope and applicability.

After the rebuttal and discussion period, we are encouraged to see consistently positive reviews, with two reviewers raising their scores. The discussion was highly productive, enabling us to address questions in detail and provide additional supporting evidence. We expanded the evaluation scope with additional model comparisons and targeted analyses, further strengthening the benchmark’s comprehensiveness and practical relevance.

We will incorporate all clarifications, supplementary analyses, and suggested improvements into the revised version to further enhance clarity, reproducibility, and the overall contribution. Once again, we are grateful for the constructive engagement and support from the reviewers and the AC, which have been invaluable in refining and improving this work.

---

### Decision · Program_Chairs · 2025-09-17

**Decision:**

Accept (poster)

**Comment:**

This paper introduces NavBench, a benchmark for evaluating multimodal LLMs in embodied navigation, with tasks covering global instruction alignment, temporal progress estimation, local reasoning, and step-wise execution under a novel difficulty-aware design. The work is well-written, and evaluates a diverse set of open-source and proprietary models, revealing important insights into temporal reasoning challenges, trade-offs between comprehension and execution, and the limited benefits of chain-of-thought prompting. While some novelty concerns exist due to reliance on existing datasets and panoramic input formats rather than the increasingly standard egocentric videos, the benchmark nonetheless fills a clear gap in evaluating navigation reasoning and supports real-world deployment scenarios. During rebuttal, the authors convincingly addressed licensing, data filtering, complexity weighting, error analysis, multilingual evaluation, and extended the model comparisons, which resolved most reviewer concerns.